# Design and Control Simulation Analysis of Tender Tea Bud Picking Manipulator

**Peng Xue** [1,2,3], **Qing Li** [1,2,3,*] and **Guodong Fu** [1,2,3]

1   School of Automation, Beijing Information Science & Technology University, Beijing 100192, China; xue564335peng@163.com (P.X.); fuguodd@163.com (G.F.)
2   Ministry of Education Key Laboratory of Modern Measurement & Control Technology, Beijing 100192, China
3   Beijing Key Laboratory of High Dynamic Navigation Technology, Beijing 100192, China
*   Correspondence: liqing@bistu.edu.cn

**Abstract:** Aiming at the current complex problem of the mechanized high-quality picking of tender tea buds, this paper designs a tender tea bud-picking manipulator. In the picking process, the quality of the petiole and leaf blade of the tender tea bud is crucial, as the traditional cutting picking method destroys the cell structure of the tender tea buds, resulting in rapid oxidation of the cuts, thus losing the bright green appearance and pure taste. For this reason, this paper draws on the quality requirements of tender tea buds and traditional manual picking technology, simulating the process of the manual picking action, putting forward a 'rotary pull-up' clamping and ripping picking method, and designing the corresponding actuating structure. Using PVDF material piezoelectric thin-film sensors to detect the clamping force of the tender tea bud picking, the corresponding sensor hardware circuit is designed. In addition, the finite element analysis method is also used to carry out stress analysis on the mechanical fingers to verify the rationality of the automatic mechanism to ensure the high-quality picking of tender tea buds. In terms of the control of the manipulator, an SMC-PID control method is designed by using MATLAB/Simulink 2021 and Adam 2020 software for joint simulation. The way to control the closed-loop system angle and angular velocity error feedback is by adjusting the PID parameters, which quickly converts the sliding mode control to the sliding mode surface. The simulation results show that the SMC-PID control method proposed in this paper can meet the demand in tender tea bud picking and simultaneously has high control accuracy, response speed, and stability.

**Keywords:** tender tea bud picking; picking manipulator; rotary pull-up picking; SMC-PID

## 1. Introduction

Tender tea buds have always been popular as a pure and high-end beverage and continue to maintain a high market demand. However, in the production of tender tea buds, the picking of tender tea buds has been a pain point for the industry. At present, tender tea bud picking is primarily manual. The picking quality is high, but taking into account that tea is a seasonal product, it is challenging to accommodate the large-scale and labor-intensive recruiting necessary during the tea-picking period. Machine-picked tea mostly compounds the cutting efficiency but at low quality, as the blade cutting will lead to the destruction of the cell structure of the tea leaves, prompting the tea to undergo oxidation faster so that it loses the fresh green appearance and pure taste, and often machine picking is only suitable for bulk tea. Selection, therefore, requires a picking robot to meet the requirements of tender tea bud picking.

As the vital actuator of the robot, the manipulator needs to face the complex picking environment and picking demand. Therefore, the accuracy, stability, and rapidity of the controller have stringent requirements [1–3]. The current control methods of the robotic arm have fractional-order PID control, particle swarm optimization fuzzy PID control,

adaptive neural PD control, and other practices [4–9]. Due to the complexity of their algorithmic parameter design, computational costs, and other issues, they are mostly not applicable to the application of tea-picking machinery. At present, domestic and foreign research on tender tea buds-choosing robotd is scarce. Considering the inherent characteristics of tea shoots, the existing picking robot volume is large, it is complicated to reach into dense tea bushes, and clamping quickly damages the tea stalks, posing a greater challenge to complete the picking of tender tea buds [10–13]. Therefore, there is a need to design a picking robot. In the paper [14–16], the picking robot was redesigned to increase the flexibility and applicability of picking by combining the characteristics of the picking targets, but the use of blade-cutting picking did not meet the needs of tender tea bud picking. Hou et al. [17] designed a soft grasping bionic robot that demonstrated soft contact mechanical behavior during gentle grasping but with large errors in terms of small target grasping. Hao [18] designed a bionic four-finger picking manipulator, imitating the artificial "hand-picking" method, but due to mechanical errors, it is more difficult to hold the rootstock of the shoots accurately, and the damage to the rootstock of the nodes is more significant, which cannot meet the standard of picking tender tea buds. Motokura et al. [19] designed a bionic three-finger selecting actuator by reproducing the three-finger picking actuator; the control method is simple, but the disadvantage is that the fingertip holding force control is not precise, and the tea shoots still easily fall off in the process.

In summary, the above manipulator in picking tender tea buds, due to the improper picking method and control method, will cause damage, affecting the quality of tender tea buds and increasing the cost of picking. Therefore, this paper designs a new type of tender tea bud-picking robot and proposes a new picking method. The main innovative work of this paper is as follows:

(1) We designed a tender tea bud-picking manipulator with the use of SolidWorks 2021 software 3D modeling to verify the reasonableness of the mechanical structure. Also in this paper, in the study of the natural growth characteristics of tea and the analysis of the traditional manual picking practices based on the proposed 'rotary pull-up' clamping and ripping picking method, PVDF piezoelectric thin film sensors are applied to determine the picking clamping force to simulate the human finger in the selection of the tender tea buds.

(2) To realize the above proposed 'rotary pull-up' clamping and ripping picking method, an SMC-PID (Sliding Mode Control–PID) controller was designed and compared with the traditional SMC and PID controllers. The results show that the SMC-PID controller offers good control performance in the control of the manipulator and can meet the needs of tender tea bud picking.

## 2. Overall Structure Design of Tender Tea Bud-Picking Manipulator

### 2.1. Picking Method and Structural Design

Tender tea buds, whose fresh leaf standards are generally one bud, one bud, and one leaf, need to ensure the integrity of the leaves [20]. At present, the highest process of plucking tender tea buds is manual plucking, and the resulting tea buds have the characteristics of good quality and high quality. As shown in Figure 1, in the analysis of hand-picked tea shoot manipulation, four steps are generally required in the plucking process: (1) Clamping: the thumb and forefinger clamp the tea leaf stalks (Figure 1a). (2) Rotation: the thumb and forefinger are rotated clockwise (Figure 1b). (3) Lifting: the thumb and forefinger are lifted upwards (Figure 1c). (4) Picking is completed (Figure 1d), and the broken pieces of the plucked tea leaves are flat and smooth. Based on the inspiration of traditional manual picking methods, this paper concludes that a 'rotating and lifting' way can ensure the quality of tender tea buds.

According to the 'rotary pull-up' clamping and ripping picking method obtained from Figure 1, the design of the tender tea bud-picking manipulator model is shown in Figure 2. As the growth of the tea environment is more complex, the location of the shoots around the branches of the tea tree will block the manipulator's movement. The area of the tea shoots of different heights will be uneven, but overall there is upward growth, so the design

of the picking manipulator is similar to the shape of the long mouth poker to pull the leaves and buds effectively.

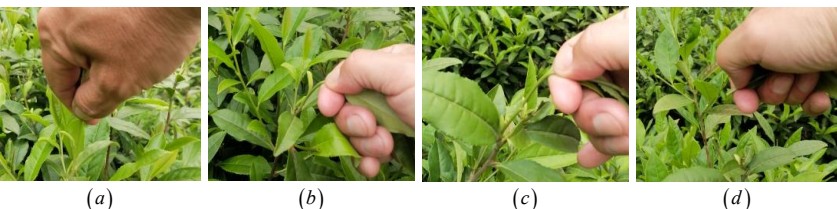

|           |           |           |           |
|:---------:|:---------:|:---------:|:---------:|
| (*a*)     | (*b*)     | (*c*)     | (*d*)     |

**Figure 1.** Schematic diagram of the decomposition of traditional manual picking of tender tea buds.

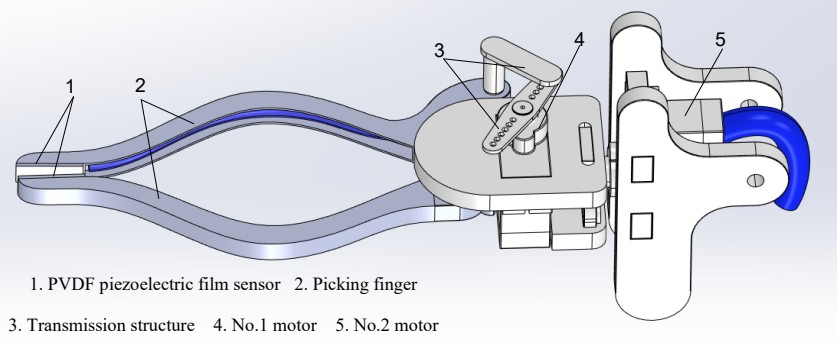

1. PVDF piezoelectric film sensor   2. Picking finger

3. Transmission structure   4. No.1 motor   5. No.2 motor

**Figure 2.** Schematic of SolidWorks modeling of a picking manipulator.

The designed manipulator for picking tender tea shoots must efficiently clip the shoots. The power unit of the picking manipulator primarily relies on an electric motor. The front-end clamping fingers are connected to the transmission device via a gear structure. The electric motor imparts power to drive the transmission device, enabling the mechanical fingers to open and close. Moreover, the inner side of the picking fingers is equipped with a PVDF piezoelectric thin film sensor. When the pressure reaches a preset value, the No. 1 motor stops functioning. The rear end is powered by the No. 2 motor, controlling the rotary action of the front-end mechanical finger part. This manipulator employs embedded alignment with the cable threaded through the inner part of the robotic finger and exiting through the tail end. This design ensures both the safety and simplicity of the manipulator's appearance during practical operation.

### 2.2. Manipulator Picking Method and Structural Design
#### 2.2.1. Working Principle

The tender tea bud-picking manipulator workflow is shown in Figure 3. In the picking, the camera collects images after target detection technology to determine the target of the tea buds, and the No. 1 motor, through the rotation of the linkage mechanism, drives the picking fingers open, close to the tea buds bud rhizome after the No. 1 motor operation, the picking fingers clamping the bud rhizome part. Because the picking finger is attached to the flexible force feedback sensor, that is, to avoid the picking process on the bud's surface and the damage's internal structure, it also can be more accurate to achieve the expected value of the clamping force. Then, the No. 2 motor starts to run, driving the whole clamping device to rotate, and stops working after reaching the preset rotation angle. Finally, the robot lifts and collects the picked tea shoots, thus completing a complete picking job.

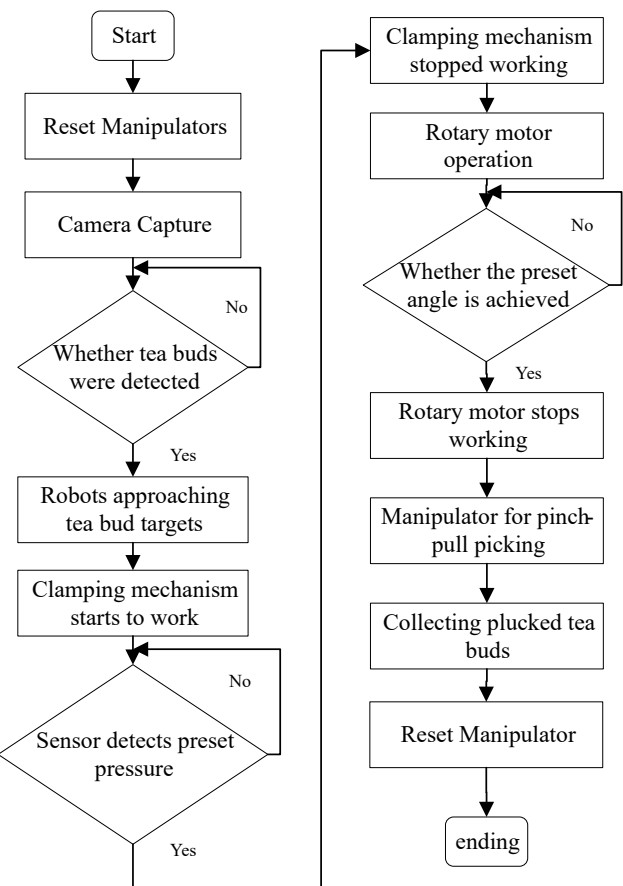

**Figure 3.** Workflow diagram of the premium tea bud-picking manipulator.

### 2.2.2. Dynamical Model

Considering the above proposed 'rotary pull-up' clamping and ripping picking method, a four-degree-of-freedom manipulator can meet the picking needs, and a four-degree-of-freedom spatial manipulator model is established, as shown in Figure 4. Assuming that the mass of the picking manipulator is uniform and friction and nonlinear effects are neglected, the length of the linkage of the robotic arm and the mass of the linkage are $L_i$ and $m_i (i = 1, 2, 3, 4)$ in Figure 4. We introduce the Lagrange equation K = T − V to determine the dynamics model of the system, where T is the kinetic energy of the manipulator and V is the potential energy of the manipulator; the kinematic equation of the manipulator can be expressed as:

$$\frac{d}{dt}\frac{\partial K}{\partial \dot{q}} - \frac{\partial K}{\partial q} = \Gamma \tag{1}$$

where $\Gamma$ represents the generalized force, since this manipulator has four linkage structures, there are four equations for the generalized coordinate $q_i$ and four equations for the generalized force $\Gamma_i$. Summing the kinetic energy of the four linkage structure links, the total kinetic energy of this manipulator is:

$$T(\theta, \dot{\theta}) \sum_{i=1}^{n} T_i(\theta, \dot{\theta}) = \frac{1}{2}\dot{\theta}^T D(\theta)\dot{\theta} \quad n = 4 \tag{2}$$

where $D(\theta) \in R^{4\times 4}$ is the inertia matrix of the manipulator. From the Lagrange equation $K = T - V$, we next require the potential energy of the manipulator arm, so we introduce $h_i$ as the height of the centre of mass of the $i^{th}$ link.

$$V_i(\theta) = m_i g h_i(\theta) \tag{3}$$

where $g$ is the Earth's gravitational acceleration and the gravitational component is:

$$g(\theta) = \frac{\partial V}{\partial \theta_i} \tag{4}$$

where the potential energy $V$ is:

$$V(\theta) = g(m_1 h_1(\theta) + m_2 h_2(\theta) + m_3 h_3(\theta) + m_4 h_4(\theta)) \tag{5}$$

where $h_i$ is the height of the centre of mass of the $i^{th}$ linkage.

$$
\begin{aligned}
h_1(\theta) &= r_1 \\
h_2(\theta) &= L_1 + r_2 \sin \theta_2 \\
h_3(\theta) &= L_1 + L_2 \sin \theta_2 + r_3 \sin(\theta_2 + \theta_3) \\
h_4(\theta) &= L_1 + L_2 \sin \theta_2 + L_3 \sin(\theta_2 + \theta_3) + r_4 \sin(\theta_2 + \theta_3 + \theta_4)
\end{aligned}
\tag{6}
$$

Finally, bringing Equations (2) and (3) into and combining with Equation (1) yields the Lagrangian dynamics equation of the manipulator as:

$$K(\theta, \dot{\theta}) = \sum_{i=1}^{n} \left( T_i(\theta, \dot{\theta}) - V_i(\theta) \right) = \frac{1}{2} \dot{\theta}^T D(\theta) \dot{\theta} - V_i(\theta) \tag{7}$$

In order to simplify the equation for ease of solution, the common expression written in matrix form is as follows.

$$D(\theta)\ddot{\theta} + C(\theta, \dot{\theta})\dot{\theta} + g(\theta) = \tau \tag{8}$$

where $D \in R^{4 \times 4}$ is the inertia matrix of the manipulator, $C \in R^{4 \times 1}$ is the Coriolis term of the manipulator, $g(\theta) \in R^{4 \times 1}$ is the vector matrix of gravity, and $\tau \in R^{4 \times 1}$ is the moment vector.

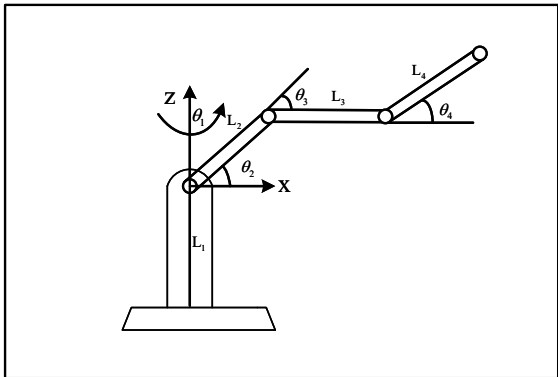

**Figure 4.** 4-DOF picking manipulator planar schematic diagram.

### 2.3. Finite Element Analysis of Picking Machine Fingers

In this study, we utilized the simulation module within SolidWorks 2021 to conduct finite element analysis of the manipulator. Given that the components in contact with tender tea buds and leaf stalks during the picking process primarily include the mechanical fingers, we focused on extracting and subjecting the picking robotic fingers to static stress analysis to validate the rationality of the mechanical structure. In this design, AISI304 stainless steel was uniformly chosen as the material for the mechanical finger, and the cylindrical aperture at the end of the mechanical finger served as the fixed geometry location point. Using the Simulation module in SolidWorks 2021 software, we performed a detailed finite element static stress analysis of the robotic finger to simulate the external load of 4 N that the manipulator experiences during the tea-picking process. By scrutinizing the stress diagram, we elucidated the stress distribution within the mechanical finger under these

working conditions. As depicted in Figure 5, the maximum stress occurs in the red region at the middle and lower end of the mechanical finger, reaching 44.95 MPa. This stress value is considerably lower than the yield strength of AISI304 stainless steel (206.8 MPa), indicating that the mechanical finger will not experience plastic deformation or damage. Additionally, as illustrated in Figure 6, the preset external load is 4 N, which is significantly below the critical yield strength of AISI304 stainless steel (18.4 N). This underscores the stability of the mechanical finger, as the external load remains well below the critical threshold. Examining the displacement in Figure 7, we observe that the maximum displacement of the robotic finger during operation occurs at the red region of the fingertip, measuring 0.1863 mm. This displacement falls within acceptable limits for the overall positioning accuracy requirement of the manipulator, indicating that the manipulator can maintain sufficient displacement accuracy during tea picking to successfully complete the task.

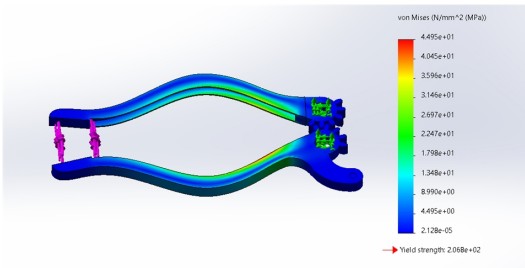

**Figure 5.** Stress cloud diagram.

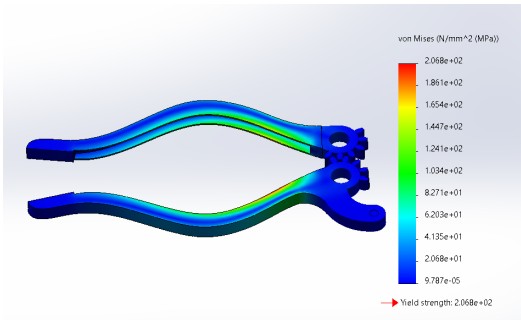

**Figure 6.** Maximum stress cloud diagram.

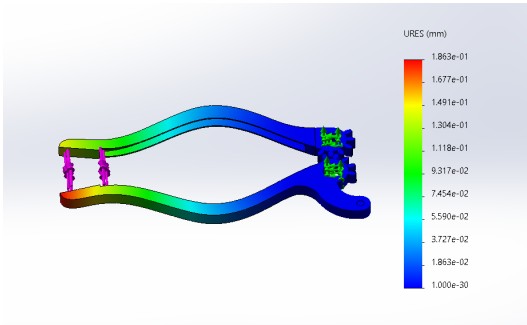

**Figure 7.** Displacement cloud diagram.

In summary, based on the comprehensive analyses conducted above, we can conclude that when the robotic finger is exposed to an external load of 4 N, both the stress and displacement of the component fall within a reasonable range. This aligns with the requirements for tea picking by the manipulator and ensures that there is no risk of material damage or performance issues. Consequently, these findings affirm the reliability and stability of the manipulator's performance.

## 3. Flexible PVDF Piezoelectric Thin Film Sensors

### 3.1. Sensor Structure and Design

To preserve the quality of tea shoots, it is imperative to maintain the force exerted by the robotic hand on the rhizome within reasonable limits. The gripping power should be as small as possible to contact with the rootstock part of the tea shoots. Leveraging the piezoelectric effect, the PVDF thin film material sensor offers significant advantages, including a substantial piezoelectric constant, excellent flexibility and elasticity, high sensitivity, and mechanical toughness. Moreover, PVDF sensors, being flexible polymer thin film elements, can be easily adhered to curved surfaces, and their flexibility and lightweight properties result in very low impedance. This characteristic enables PVDF sensors to deform under very small forces [21–23]. In this study, an LDTI-028K PVDF piezoelectric film sensor is employed to detect the clamping force, and the typical performance parameters of its piezoelectric film are presented in Table 1.

**Table 1.** Typical performance parameters of PVDF.

| Thickness (mm) | Piezoelectric Stress Constant (p.CN-1) | | | Density $(g \cdot cm^{-3})$ | Young's Modulus (Gpa) |
|---|---|---|---|---|---|
| | d31 | d32 | d33 | | |
| 55 | 23 | 3 | −33 | 1.78 | 2–4 |

The sensor incorporates a 28 μm piezoelectric film with screen-printed silver paste electrodes. This film is laminated onto a 0.125 mm polyester substrate, and two crimp terminals guide the electrodes. When subjected to vibration conditions, the PVDF piezoelectric film undergoes bending movements, causing membrane elongation and shortening. These deformations generate tensile and compressive stresses, consequently producing corresponding voltage outputs. The membrane structure of the PVDF and the cut-out structure are illustrated in Figures 8 and 9.

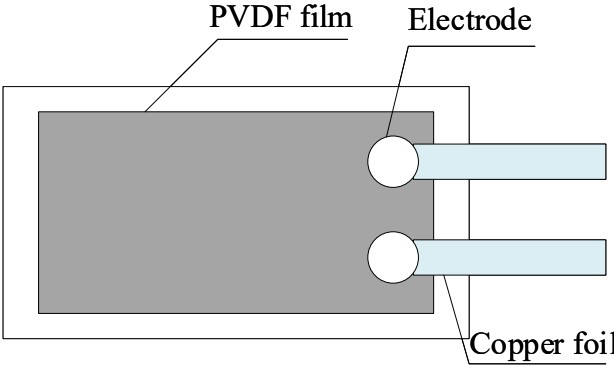

**Figure 8.** Structure of PVDF film.

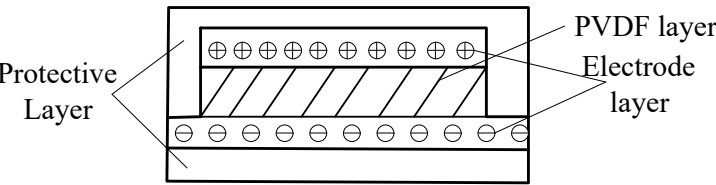

**Figure 9.** PVDF cut-out structure.

According to the literature [18], actual measurements indicate that the average length of the second pitch of the tea stalk to be grasped falls within the range of 8–12 mm. Consequently, in this study, the mechanical finger is designed with a width of 8 mm and a length of 20 mm. The dimensions of the inner-cover sensor are set at 6 mm × 16 mm, ensuring

optimal contact between the sensitive element and the tender tea buds and rhizomes. The picking finger adopts a hollow structure internally, facilitating cable organization and ensuring picking safety. The structure is depicted in Figure 10.

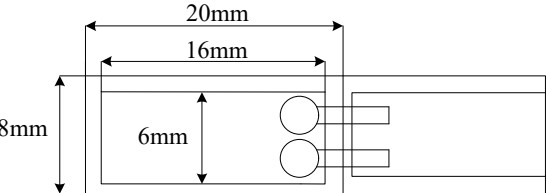

**Figure 10.** Dimensioning drawing of PVDF sensor.

### 3.2. Hardware Circuit Design

To address the challenge of the relatively small amplitude of surface charge generated by PVDF piezoelectric films when exposed to external forces, especially in the vibration-prone environment of tea-picking operations, a signal conditioning circuit has been proposed. The aim is to amplify the amplitude of the sensor output signal and minimize noise levels, thereby enhancing the signal-to-noise ratio. The adjustment circuit comprises a pre-amplification module and a low-pass filter module designed to amplify and condition the charge signal produced by the sensor. It is noteworthy that to mitigate stability and noise issues associated with traditional op-amps, resistors, and capacitors, we have embraced a functional integrated chip design. This design incorporates components such as the VK10x and the UAF42, as illustrated in Figure 11. Through this approach, we have successfully enhanced the system's performance, rendering it more suitable for complex operating environments.

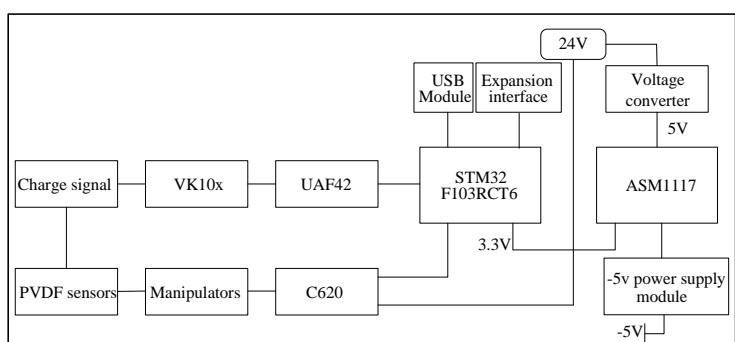

**Figure 11.** Signal acquisition and processing hardware circuit structure.

In this configuration, a pre-amplification module has been devised using the charge amplifier VK10x (NGK, Baltimore, MD, USA). The primary function of this module is to convert the high-impedance charge signal produced by the PVDF piezoelectric membrane into a low-impedance voltage signal, effectively amplifying the sensor output signal. In this process, to mitigate the adverse effects of high-frequency noise on system performance, we incorporated the UAF42 (Texas Instruments, Dallas, TX, USA) active filtering module for signal filtering, which was set at a filtering frequency of 50 Hz. The specific circuit design is illustrated in Figure 11. The processing module features the STM32F103RCT6 microcontroller (STMicroelectronics, Geneva, Switzerland) from Efate Semiconductor. This microcontroller integrates an A/D conversion function, enabling the rapid conversion of voltage analog signals into digital signals. Additionally, the processing module is equipped with a USB communication module, a program download port, and a crystal module to facilitate data transmission and program burning, ensuring the normal operation of the microcontroller. The input voltage of the hardware circuit of the system is 24 V. To ensure the normal operation of the modules, we use a voltage converter module to regulate the supply voltage. Firstly, the 24 V input voltage is stepped down to 5 V, and then the 5 V

voltage is further converted to 3.3 V by an ASM1117-3.3 (UMW, Washington, DC, USA) forward regulator for stable operation of the microcontroller. Since the VK10x and UAF42 chips require a −5 V power supply, we also added a −5 V power supply module to ensure the proper operation of the chips.

### 3.3. Simulation Results and Analysis

The parameters of the sensor employed in this study are detailed as follows: the sensitivity of the clamping force is 0.5635 V/N, with a linearity of 4.84%, accuracy of 6.41%, hysteresis of 3.53%, and a range spanning 0–8 N. These parameter settings have been incorporated into the model for simulating clamping experiments, wherein a cylinder is utilized to represent the rhizomes of tender tea buds. The simulation experiments of the manipulator grasping are illustrated in Figure 12.

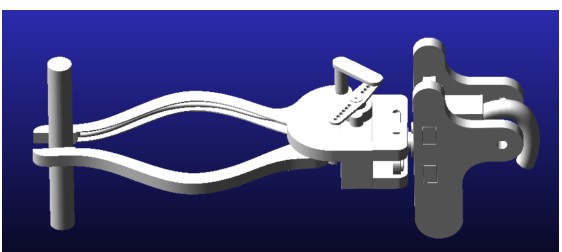

**Figure 12.** Schematic diagram of clamping simulation.

The magnitude of the clamping force selected in the simulation experiment was determined by averaging actual measurements from 120 sample data points obtained from the literature [18]. Based on this, the range of tensile forces for lifting and pulling the shoot without slipping was set between 3 and 7 N. Analyzing the response curve of the clamping force measured in the simulation experiments, depicted in Figure 13, the manipulator ideally maintains a clamping force between 3.76 and 4.43 N within a timeframe of 0.08 s. This clamping force range encompasses the force necessary to prevent slippage and avoid clamping breakage during the clamping process, thereby meeting the harvesting clamping force requirements. Therefore, the incorporation of PVDF piezoelectric thin-film sensors into the tea-picking manipulator is highly suitable. Their high measurement sensitivity, good flexibility, and low cost provide valuable insights for perceiving the clamping force of the tea-picking manipulator and enabling feedback adjustment of the clamping force. This approach fully meets the requirements for measuring the clamping force of the picking manipulator.

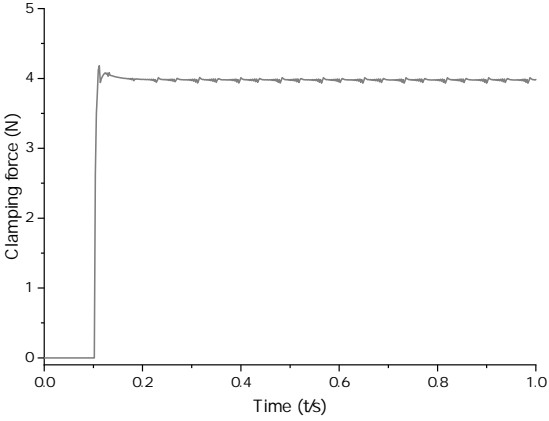

**Figure 13.** Schematic diagram for model an SMC-PID control system in Simulink.

## 4. Simulation Results and Analysis

Given the necessity for the picking manipulator to operate in a complex environment, dealing with variables like fluctuations in tea leaf growth, wind, and slope changes, the precise control of the position, speed, and force of the tea-picking manipulator is crucial. This precision control ensures accurate picking operations, ultimately enhancing the quality and efficiency of tea picking. This paper introduces a combined approach, employing sliding mode control (SMC) and PID control, which is known as the SMC-PID controller. This controller is designed to manage the robot's rotation and lifting during the picking operation, providing high-precision control over position, speed, and force. The SMC-PID controller offers the capability to adapt to the dynamic changes in tea leaf growth and the uncertainties in the picking environment. It exhibits fast response and dynamic adjustment abilities along with anti-interference properties and robustness. This design is effective in handling external interferences, contributing to the overall reliability and stability of the tea-picking manipulator in challenging working conditions.

### 4.1. Control System Modeling and Stability Analysis

4.1.1. Modeling of SMC-PID Control System

For dynamic modeling of the picking manipulator joints, the relationship between moment and acceleration is as follows:

$$\Gamma = J\ddot{\theta} \tag{9}$$

where $\Gamma$ is the moment, $J$ is the rotational inertia of the rotating joint, $\ddot{\theta}$ is the angular acceleration of the rotating joint,

The manipulator joint rotation deviation angle $e$ is:

$$e = \theta - \theta_o \tag{10}$$

where $\theta$ is the desired rotation angle, $\theta_o$ is the actual rotation angle, and the above Equation (9) can be obtained by taking the first and second derivatives:

$$\dot{e} = \dot{\theta} - \dot{\theta}_o$$
$$\ddot{e} = \ddot{\theta} - \ddot{\theta}_o \tag{11}$$

where $\dot{e}$ is the rotational deviation angular velocity, $\ddot{e}$ is the rotational deviation angular acceleration, $\dot{\theta}$ is the desired rotational angular velocity, $\ddot{\theta}$ is the desired rotational angular acceleration, $\dot{\theta}_o$ is the actual rotational angular velocity, and $\ddot{\theta}_o$ is the actual rotational angular acceleration, when the manipulator structure is in an initial equilibrium condition $\dot{\theta}_o = 0, \ddot{\theta}_o = 0$. Therefore:

$$\dot{e} = \dot{\theta}, \ddot{e} = \ddot{\theta} \tag{12}$$

Design of slide mode control for slide mode surfaces $s$:

$$s = ce + \dot{e} \tag{13}$$

The derivation of this leads to equation:

$$\dot{s} = c\dot{e} + \ddot{e} \tag{14}$$

Bringing Equation (14) into Equation (9), the control torque equation for the sliding mode controller can be obtained:

$$\Gamma = J(\dot{s} - c\dot{e}) \tag{15}$$

An exponential convergence rate is chosen, which is given by the following formula:

$$\dot{s} = -\lambda \operatorname{sgn}(s) - ks \quad \lambda > 0, k > 0 \tag{16}$$

Among them:

$$\text{sgn}(s) = \begin{cases} s > 0, \text{sgn}(s) = 1 \\ s = 0, \text{sgn}(s) = 0 \\ s < 0, \text{sgn}(s) = -1 \end{cases} \tag{17}$$

where $\text{sgn}(s)$ is a symbolic function.

Based on the above derivation, Equation (16) can be brought into Equation (15) to obtain the manipulator joint control torque as:

$$\Gamma = J(-\lambda \, \text{sgn}(s) - ks - c\dot{e}) \tag{18}$$

Utilizing Equation (9) as the basis for the sliding mode control structure in rotation, traditional sliding mode control encounters challenges during motor output control, leading to violent jitter and limitations in achieving high-precision control. To address this issue, a hybrid approach known as the SMC-PID control method is designed. This method integrates PID controllers to handle the system's error signals, allowing the sliding mode controller to directly influence the system state changes. This hybrid approach effectively mitigates the jittery control torque situation, enabling precise control of the system:

We design the sliding mould surface using the PID parameter design as follows:

$$s = K_p e + K_i \int_0^t e\,dt + K_d\dot{e} \qquad K_p, K_i, K_d > 0 \tag{19}$$

Its derivation is shown below:

$$\dot{s} = K_p\dot{e} + K_i e + K_d\ddot{e} \tag{20}$$

Bringing the above Equation (20) into Equation (18), then the joint control moments of the manipulator system can be obtained as follows:

$$\begin{aligned} \Gamma &= \frac{J}{K_d}(\dot{s} - K_p\dot{e} - K_i e) = \\ &\frac{J}{K_d}(-\lambda \, \text{sgn} - ks - K_p\dot{e} - K_i e) \qquad \lambda > 0, k > 0 \end{aligned} \tag{21}$$

where the specific parameter settings in the SMC-PID control are shown in Table 2.

**Table 2.** Controller adjusts time and overshoot comparison table.

| SMC-PID Parameter | $k_p$ | $k_i$ | $k_d$ | $\lambda$ | $k$ | Gain |
|---|---|---|---|---|---|---|
| Rotate 30° | 0.22 | 0.0001 | 0.03 | 0.05 | 400 | 1 |
| Lift 45° | 0.01 | 0.001 | 0.0175 | 0.1 | 5 | 3000 |

4.1.2. Liapunov Stability Analysis

The Lyapunov function is chosen for its system stability proof using the Lyapunov function:

$$V = \frac{1}{2}s^2 \tag{22}$$

If, at the equilibrium point $s = 0$ , the function satisfies the following conditions, then it meets the Lyapunov function condition. $s$ will eventually stabilize at the slip-mold surface, and the equilibrium is a steady state.

$$\begin{aligned} &\lim_{|s|\to\infty} V = \infty \\ &\dot{V} < 0, s \neq 0 \end{aligned} \tag{23}$$

It is clear that the continuous function is the first of the above conditions to be met for the second condition:

$$\dot{V} = s\dot{s} = s(-\lambda \operatorname{sgn} - ks) \qquad \lambda > 0, k > 0 \tag{24}$$

Therefore, when $s > 0$, $-\lambda \operatorname{sgn} < 0$, $-ks < 0$, $\dot{V} < 0$. So, the SMC-PID control system satisfies the Lyapunov stability condition and is able to reach the equilibrium state. Based on the principles of the sliding mode controller, it is evident that when the system reaches a stable state, i.e., when it enters the sliding mode surface, the error consistently converges to zero exponentially. This manifests as a strong tracking ability of the system's output to the input, demonstrating robustness in the face of disturbances [24].

*4.2. Switching Function Design*

According to the characteristics of PID control and sliding mode control, this paper combines the two controls to achieve the reduction of the impact of mechanical shock when switching from manipulator clamping operation to rotary operation, and it improves the response speed from the occurrence of mechanical shock back to the original clamping force. Assuming that $U_{pid}$ represents the PID control, $U_{smc}$ represents the SMC control, $w$ represents the clamping force fed back from the system, and $\phi$ represents the coefficients of the switching function $r$, the switching function is designed in Simulink as follows:

$$r = \phi U_{pid} + (1 - \phi)U_{smc} \tag{25}$$

$$\phi = \begin{cases} 0, w < 0 \\ 1, w \geq 0 \end{cases} \tag{26}$$

**5. Experiments and Results**

The joint simulation was carried out in MATLAB/Simulink 2021 and the Adams 2020 software environment, and the simulation model was built according to the principle of the SMC-PID control system. This experimental simulation adopts Longjing No. 43 famous tea as the simulation object for analysis. During the picking process, a 'rotary pull-up' clamping and ripping picking method was proposed to account for the individual differences in tea shoots. This method ensures that a sufficiently large force is applied to clamp the root and stem parts of the tea shoots, preventing slipping or bias. Simultaneously, the clamping force must not be excessively high to avoid causing harm to the nodes. After reviewing the literature [18,25], it can be seen that in order for Longjing 43 tea to maintain tender tea buds in the picking process, the pull-off force must be 3–7 N to avoid slippage, and the shoot rhizomes need to be clamped off within the clamping force range of 4–8 N. After careful consideration of the characteristics of the tea, this paper adopts the value of the clamping force for the picking of the simulation experiments with a value of 4 N. In this paper, the MATLAB/Simulink 2021 joint Adams 2020 joint simulation model system is shown in Figure 14.

The system design input expectation value is all step signal, the picking rotation angle is set at 30°, the lifting angle is set at 45°, the total time of the overall picking process is set at 1.6 s, the initial speed of the system is given as 0, and the difference between the angle feedback of the system and the speed feedback is input into the SMC control system as a parameter of the PID, which enables us to achieve the optimization of the SMC control system by eliminating the phenomenon of jitter and improving the speed and accuracy of the control.

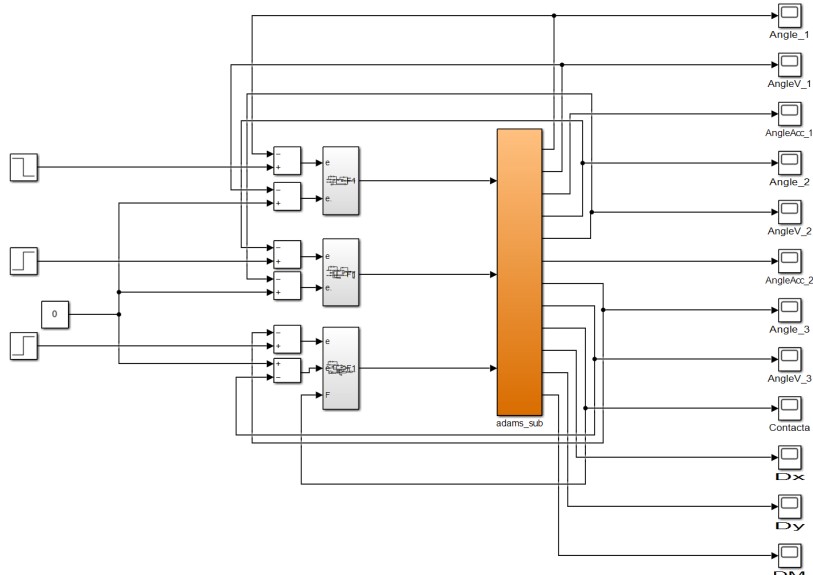

**Figure 14.** Schematic diagram for modeling an SMC-PID control system in Simulink.

In the design of the mechanical finger for clamping tea buds' rhizomes, the goal is to maintain the clamping force at 4 N to meet picking requirements. However, due to the inability to derive the derivatives of the robot force, the SMC method cannot be utilized. Additionally, in the PID control, at 0.7 s, the clamping force jumps to 0 N, as shown in Figure 15, which is followed by the recovery of the clamping force to 4 N within 0.055 s. To optimize the control effect, a composite approach is adopted where PID and SMC controls are combined. The SMC control is applied before the target is clamped, and after the clamping, it switches to PID control. In this optimized scenario, the clamping force jumps to 1.15 N at 0.7 s, and then rapidly recovers to 4 N within 0.015 s. The composite control approach demonstrates faster and more reliable convergence compared to using PID control alone.

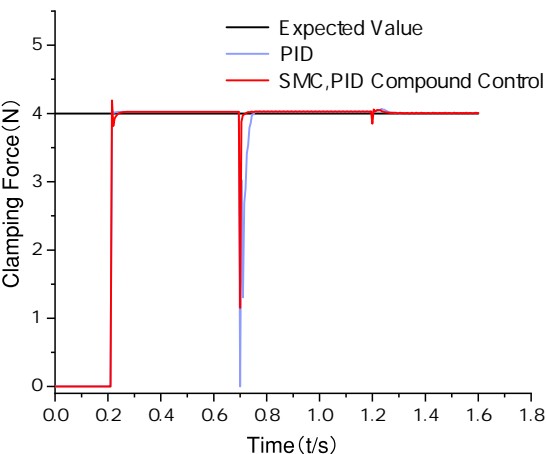

**Figure 15.** Comparison diagram of picking manipulator clamping force control.

From Figures 16 and 17, and Table 3, it is evident that the sliding mode variable structure control of SMC is prone to issues such as jitter, vibration, and hysteresis. This results in control accuracy that is challenging to meet the actual picking requirements. On the other hand, PID control tends to exhibit excessive overshooting, and its control speed and accuracy are insufficient to fulfill the demands of picking tender tea buds. However, the SMC-PID control shows superior stability compared to separate SMC and PID controls, as demonstrated in Figure 16 for rotary angle control. The SMC-PID control

curve is smoother than the PID and SMC control curves, and there is no evident vibration phenomenon observed at 0.7–0.8 s. During the transition to the lifting operation at 1.2 s, the jitter recovery is more rapid in the SMC-PID control, taking only 0.02 s, while PID control takes 0.1 s. Table 3 reveals that the overshooting amount of the rotational angle control of the manipulator is 1.38% for SMC-PID and 3.37% for PID control alone. SMC-PID reduces the overshoot amount by 59.05% compared to PID control alone. In Figure 17, for the lifting angle control, SMC-PID demonstrates greater accuracy compared to traditional SMC and PID control. The overshooting amount for PID control is 11.6%, whereas SMC-PID exhibits no obvious overshooting with the shortest adjustment time being 1.28 s. This indicates that SMC-PID achieves more precise control with minimal overshooting during the lifting and pulling operation.

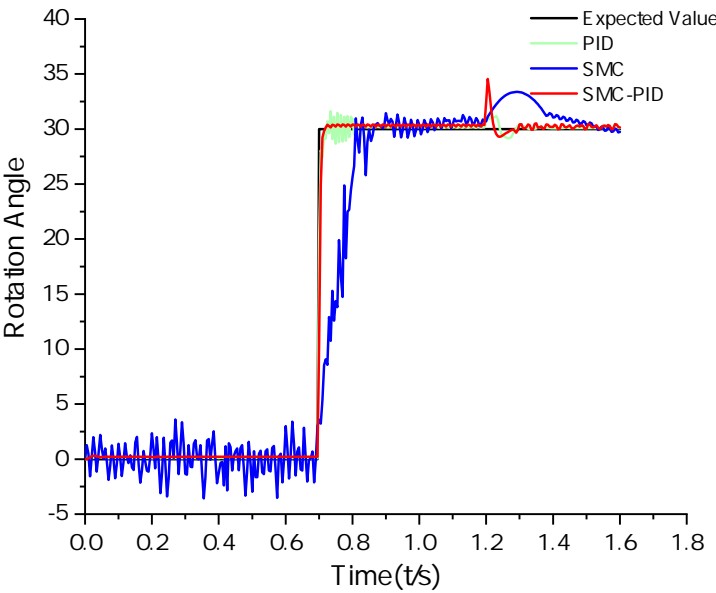

**Figure 16.** Comparison of the effect of picking manipulator lifting 30° control.

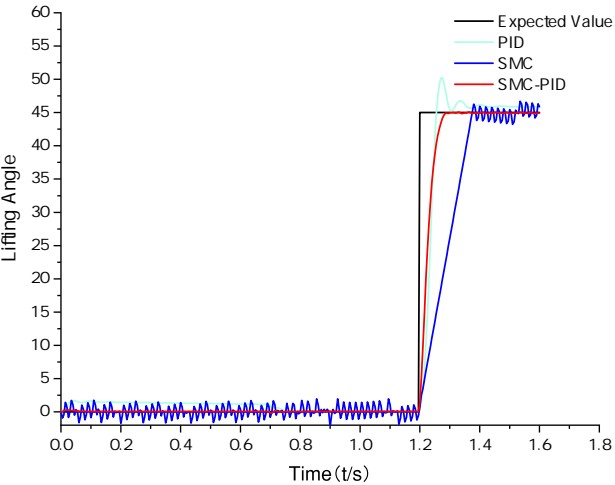

**Figure 17.** Comparison of the effect of lifting 45° control of the picking manipulator.

**Table 3.** Controller adjusts time and overshoot comparison table.

| | Controllers | Rotary Adjustment (Time/s) | Lift Adjustment (Time/s) | Rotational Overshoot (%) | Lift Overshoot (%) |
|---|---|---|---|---|---|
| 1 | PID | 0.76 | 1.42 | 3.37 | 11.6 |
| 2 | SMC | \ | \ | \ | \ |
| 3 | SMC-PID | 0.715 | 1.28 | 1.38 | 0 |

In Figures 18 and 19, upon comparing the angular velocity curves of the three control algorithms, it is evident that the peak value of the angular velocity is the smallest in SMC-PID, and the angular velocity curve returns to a smooth state more quickly. Therefore, SMC-PID control exhibits superior motion smoothness, indicating that the robot can reduce swing and jitter during the motion process, making the robot's movement smoother and more stable. This observation emphasizes that SMC-PID control is softer, more accurate, and more reliable, with no overshoot. It proves to be better suited for satisfying the demand of picking tender tea buds compared to traditional SMC and PID control methods.

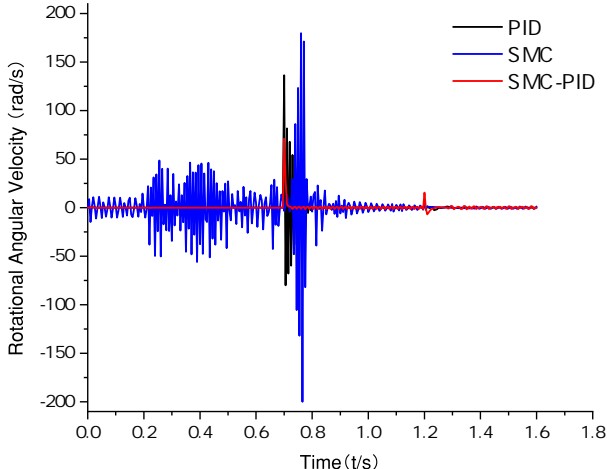

**Figure 18.** Comparison of angular velocity control for a 30° rotation of a picking manipulator.

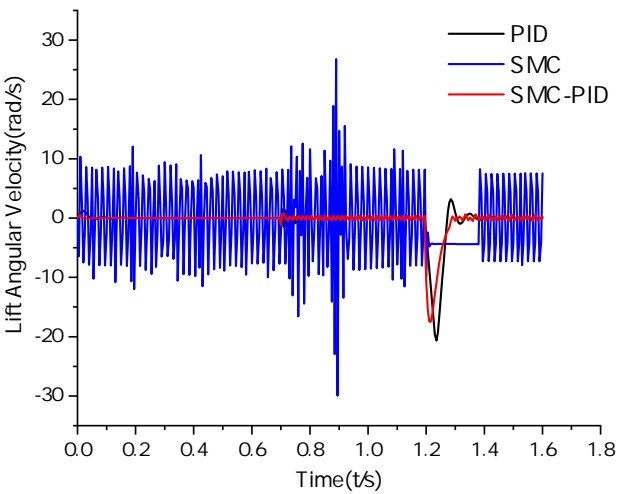

**Figure 19.** Comparison of angular velocity control of a picking manipulator lifting at 45°.

In the picking process, to ensure the smooth picking of tender tea buds, it is essential to measure the lifting distance of the picking robot. As shown in Figures 20 and 21, the X-axis lifting distance is approximately 37.7 mm, and the Y-axis lifting distance is about 107.5 mm. Additionally, Figure 22 illustrates that the combined displacement of the manipulator is

approximately 113.9 mm. This lifting distance is entirely sufficient to meet the requirements of standard picking.

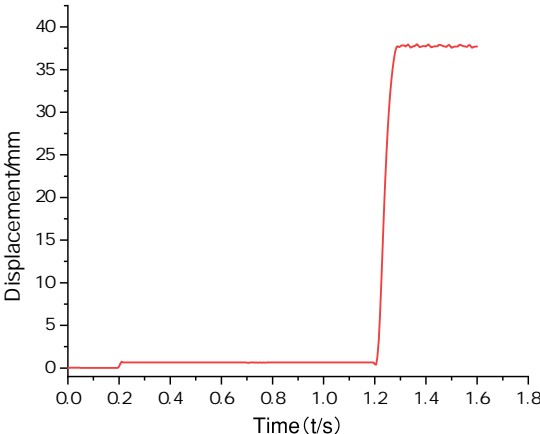

**Figure 20.** Displacement distance of the X-axis of the manipulator.

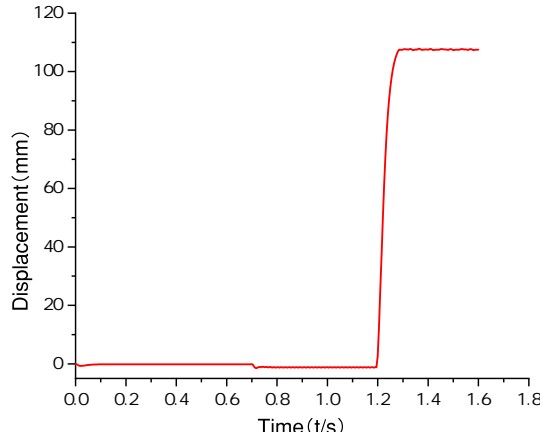

**Figure 21.** Displacement distance of the Y-axis of the manipulator.

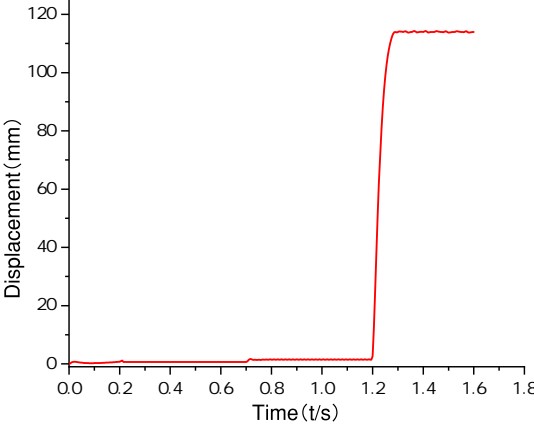

**Figure 22.** Manipulator and displacement distance.

## 6. Conclusions

(1) In this study, the growth characteristics of tender tea buds and traditional manual picking methods were analyzed, leading to the proposal of a 'rotary pull-up' clamping and pulling picking method. This method simulates the non-destructive manual picking of tender tea buds, and a corresponding picking robot for tender tea buds shoots was designed.

A manipulator model was established in SolidWorks 2021, and its mechanical structure was verified through finite element analysis. Additionally, a PVDF piezoelectric thin film sensor was introduced and mounted on the mechanical finger for simulation experiments. The results of the experiments confirmed the rationality of the mechanical structure, and the introduction of the PVDF piezoelectric thin film sensor successfully validated the manipulator's ability to meet design requirements. It demonstrated the capability to effectively reach into the complex growth environment of tea leaves for picking.

(2) Joint simulation is conducted in MATLAB/Simulink 2021 and Adams 2020 software. The SMC-PID controller is designed in Simulink, enabling the control of the closed-loop system's angular and angular velocity error feedback by adjusting PID parameters. This adjustment facilitates the quick convergence of sliding mode control to the sliding mode surface. The regulation time and overshoot are significantly reduced in comparison with traditional PID and SMC control methods. In the control of the clamping force, the composite control method effectively mitigates the jump phenomenon of the clamping force during the picking process. Simulation results demonstrate that the SMC-PID controller exhibits higher control accuracy, stability, and response speed when compared to traditional PID and SMC control methods.

**Author Contributions:** Methodology, P.X. and Q.L.; Validation, P.X.; Resources, Q.L.; Data curation, G.F.; Writing—original draft, P.X.; Writing—review & editing, G.F.; Project administration, Q.L. and G.F. All authors have read and agreed to the published version of the manuscript.

**Funding:** This research was funded by The National Key Research and Development Program of China grant number 2022YFF0607403 and was funded by Supported by The National Natural Science Foundation of China grant number 61971048.

**Institutional Review Board Statement:** Not applicable.

**Informed Consent Statement:** Not applicable.

**Data Availability Statement:** Data are contained within the article.

**Conflicts of Interest:** The authors declare no conflicts of interest.

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
