# Peer review of "Design and Control Simulation Analysis of Tender Tea Bud Picking Manipulator"

_applsci, doi:10.3390/app14020928_

Round 1
Reviewer 1 Report
Comments and Suggestions for Authors
The paper deals with the design of a tender tea bud-picking manipulator and the picking clamp. In the work, the authors detail the characteristics of the clamp and the control effort to avoid damaging the tender tea bud.
The paper is well written, and the presentation of the clamp is wide from a scientific point of view.
The reviewer suggests that the scientific presentation of the paper can be improved by considering a spatial robot instead of a planar one, as proposed by the authors from line 115 to line 140.
The authors can introduce a general formulation of the spatial robot that uses the special picking clamp, focusing the paper on the picking problem.
Reviewer 2 Report
Comments and Suggestions for Authors
The manuscript presents an innovative approach to automating the process of picking tender tea buds. The authors propose a novel 'Rotary pull-up' clamping and ripping picking method, coupled with a specially designed manipulator. The work integrates advanced technologies such as PVDF piezoelectric thin-film sensors, MATLAB/Simulink, and Adams software for control simulation, and finite element analysis for stress testing.
Clarity and Organization: The manuscript is dense and could benefit from a more structured layout. Subheadings, bullet points, and clearer separation of sections would enhance readability.
Comparative Analysis: While the proposed method is novel, a comparative analysis with existing technologies or methods in the field would provide a better understanding of its advantages and limitations.
Economic and Practical Feasibility: The manuscript lacks discussion on the economic feasibility and practical implementation challenges of deploying such a system in real-world tea plantations.
Long-Term Durability Testing: There is a need for long-term durability testing of the manipulator in real field conditions to assess its performance over time.
The use of technical jargon is heavy; a glossary of terms or simpler explanations might be beneficial for a broader audience.
The manuscript presents significant advancements in the field of agricultural robotics, specifically in the context of tea bud picking. However, the addition of comparative studies, economic analysis, and field-testing data would greatly enhance the value of the research.
Comments on the Quality of English Language
Some typographical and grammatical errors need correction for professional presentation.
Reviewer 3 Report
Comments and Suggestions for Authors
The paper presents a study on mechanical design and control of a gripper for tea picking. The authors propose a mechanical design created in SolidWorks, design of electronic system and a hybrid controller combining PID and slide-mode control. The controller is then tested in Matlab simulations.
The problem of automated tea picking undertaken by the authors is interesting and a potential solution to it may have a practical impact to the industry. Yet the paper seems to use the "tea picking" only as a background and the actual results presented are not specific enough, not considering the unique problems related with the claimed application. The results are limited to simulations and with the number of simplifying assumptions seems not sufficient. Moreover the authors do not provide an extensive analysis to the problem that would justify the parameters for the simulations they present (for example the simulations for clamping motion in 2.2 show the clamping force stabilized within 3.76-4.43N in 0.8s, what is summarized by the authors as "fully meeting expected demand" - yet those demands were not specified in the paper; similar situation is with other parameters and conclusions).
In the details of the parts of the text:
1) Mechanical design - that part seems to be the most complete part of the paper, presenting both the design and the mechanical analysis. The presented simulations are yet limited to a single grasping of a vertical rigid cyllinder of unknown diameter measuring clamping force and clamp stabilization time. It is not possible for the reader to evaluate how realistic are the parameters of those simulations. The problems of various diameters or flexibility of tea-buds are not discussed, neither are problems of possible slipping, influence of plucking moment or multiple stalks within the gripper clamps.
2) Electronic design - that part seems in this paper as only a proposition of the connections in a block diagram and white-paper based parameter check is provided without practical influence on the rest of the paper. For current content only the force sensor parameters are relevant and only those should remain in the paper until a real device is built.
3) The design of control algorithm uses a combination of 2 standard approaches: PID and slide-mode. It is not clear how the parameters of the algortihms were selected therefore it is not clear if the better performance of the proposed solution is the result of objectively better algorithm or just a specific choice of the parameters.
Additional remarks:
a) some parts of the text seems redundant, specifically:
- general derivation of standard dynamic equations given in details - it would be enough to put in the text a reference and the formulas specific for the considered problem
- general derivation of slide-mode control - as above
- hardware circuit design - only the sensor parameters are useful for the rest of the text,
b) the equations of the dynamics seem to be oversimplified as the authors derive them for a planar manipulator moving in a vertical plane, while the later used motion uses different angles (and therefore changing the influence of the gravity force to the joints)
c) the readability of all remaining mathematical formulas should be improved, with an attention to details - the most visible issue is in (19) - I expect the authors di not mean a product of Kp Kd Ki>0, but each of the parameters to be positive - but apart of that a number of other formulas require similar editing to improve their readability.
Round 2
Reviewer 3 Report
Comments and Suggestions for Authors
The authors reply to my previous comments and the changes introduced to the revised version of the paper clarify most of the issues.
General comment:
- the authors addressed the issue of planar manipulator raised by the reviewers by proposing the structure from Fig. 4; that however solves the issue only partially as it does not seem to correspond to the picking motion proposed by the authors - the mechanical structure and the algorithm define picking as "roll" rotation in the last joint, but the structure in Fig. allows only yaw and pitch rotations.
- I maintain may opinion about the section 2.2 - it has too many details as for presenting an idea, but too few details to present a solution; if the authors want this section as presenting a design - what naming of each single component suggest - I would expect as detailed analysis as is provided for the mechanical design (and I would have more questions on the requeirements and the motivation for each component choice supported by analysis); however if authors want only to present the idea, a block named "voltage converter" would be more appropriate in place of "LM2596s" (and more comprehensible to the reader who wants to understand how the system work and not to build it).
Minor remarks regarding editing:
- Fig.4 caption - it is not 2-dof manipulator any more
- K appearing in eq. (1) is not defined at that moment and is only defined later in l. 129; it would be better to move the explanation of K=T-V around l. 121 where T and V are defined
- The authors repeatedly use the term "slide mould" for what is known in literature as "slide (or sliding) mode".
- I maintain my remark that in (19) the condition should be "K_p, K_i, K-d >0" not "K_p K_i K-d >=0" (two issues here: 1) commas, as each of the parameters must be positive not only their product, 2) strict inequality, as using >= allows all of the parameters to be 0 what certainly will lead to failure of the controller)
- I suggest to use some space in formulas to separate the main formula from the conditions, for example: in (19) after the comma; in (20) after the first comma, before "\lambda>0" - simply in all equations containing in a single line a formula and conditions for variables
- l. 120 missing spaces around "and" (with the above remark - probably an issue with horizontal spaces in LaTeX math mode)
- the used form of presentation of ranges as for exmple "[3-7N]" seems unusual, I would expect just 3--7N (with -- being a long dash); I have noticed it in lines: 271,361,362, possibly also elsewhere
- the font choice for plots in Figs.15-22 makes the axes labels and especially legends (when present) hard to read - I suggest to use sans serif font of bigger size
